# Social and endogenous infant vocalizations

**Helen L. Long**[1]*, **Dale D. Bowman**[2], **Hyunjoo Yoo**[3], **Megan M. Burkhardt-Reed**[1], **Edina R. Bene**[1], **D. Kimbrough Oller**[1,4,5]

**1** Origins of Language Laboratory, School of Communication Sciences and Disorders, University of Memphis, Memphis, Tennessee, United States of America, **2** Department of Mathematics, University of Memphis, Memphis, Tennessee, United States of America, **3** Department of Communicative Disorders, College of Arts & Sciences, The University of Alabama, Tuscaloosa, Alabama, United States of America, **4** Institute for Intelligent Systems, University of Memphis, Memphis, Tennessee, United States of America, **5** Konrad Lorenz Institute for Evolution and Cognition Research, Klosterneuburg, Austria

* hlong@memphis.edu

**Data Availability Statement:** All data required to replicate the study's findings are contained in the manuscript, its files, and openly available in Open Science Framework at https://osf.io/hb834/. Due to the nature of this research, participants of this

## Abstract

Research on infant vocal development has provided notable insights into vocal interaction with caregivers, elucidating growth in foundations for language through parental elicitation and reaction to vocalizations. A role for infant vocalizations produced endogenously, potentially providing raw material for interaction and a basis for growth in the vocal capacity itself, has received less attention. We report that in laboratory recordings of infants and their parents, the bulk of infant speech-like vocalizations, or "protophones", were directed toward no one and instead appeared to be generated endogenously, mostly in exploration of vocal abilities. The tendency to predominantly produce protophones without directing them to others occurred both during periods when parents were instructed to interact with their infants *and* during periods when parents were occupied with an interviewer, with the infants in the room. The results emphasize the infant as an agent in vocal learning, even when not interacting socially and suggest an enhanced perspective on foundations for vocal language.

## Introduction

### Overview

The relative frequencies of human infant vocalizations that can be categorized as social vs. endogenous have not been a major focus of research. We seek to quantify the extent to which infants vocalize socially and endogenously in naturalistic settings. The effort has led to a shift in our perspective, where the contribution of endogenous vocalization and exploratory vocal play has assumed increasing importance in our speculations about the emergence of the speech capacity both in development and evolution.

The new perspective is informed by evolutionary developmental biology, evo-devo [1–4], a paradigm of thought that emphasizes natural selection as targeting developmental processes, allowing the evolution of foundational structures and capabilities upon which subsequent developments can self-organize and be further exploited in subsequent development and evolution. This approach does not diminish the importance of social interaction in the origin of the speech capacity, but instead is hoped to help account for foundational requirements of

study did not agree for audio/video recordings to be shared publicly, so raw recording data is not available, nor required to replicate the study.

**Funding:** The research for this manuscript was funded to DKO by the National Institutes of Health grants R01 DC006099, DC011027, and DC015108 from the National Institute on Deafness and Other Communication Disorders (https://www.nidcd.nih. gov/) and by the Plough Foundation (http://plough. org/) which supports DKO's Chair of Excellence. The funders had no role in study design, data collection and analysis, decision to publish, or preparation of the manuscript.

**Competing interests:** The authors have declared that no competing interests exist.

functionally flexible vocal interaction. In essence, the line of reasoning emphasizes the origin of flexible vocalization, without which significant growth in flexible vocal interaction and, through further development, vocal language may have been impossible.

## Social interaction and vocal development

The effect of social interaction on infant vocal development has long been a topic of interest in child psychology and the emergence of language [5–13]. The study of infant intrinsic motivation for social engagement has highlighted an apparently innate drive to engage in face-to-face dyadic interaction with caregivers from birth [14, 15] and has been interpreted as contributing to the development of temporal sensitivity, vocal coordination, and social contingency [16–20]. The long tradition of research in infant attachment and bonding [21–24] has included a distinct emphasis on the parent-infant dyad as the fundamental unit of human social and emotional development. Even in the first 3 months of life parent-infant vocal interaction has been described in detail [25–27]. Experimental studies in the still-face paradigm [28] have shown that by 5–6 months of age, infants increase their rate of speech-like vocalizations when the parent disengages from an ongoing vocal interaction [29, 30], suggesting infants by that age seek to repair broken interactions with increased vocalization. A social feedback loop has been posited to exist in infant and child vocalization, and that loop has been thought to promote contingent infant vocalizations with respect to caregiver vocalizations [6, 31–33]. Winnicott [34] went so far as to say that "there is no such thing as an infant," highlighting the idea that without a mother, an infant cannot exist. But this idea has been taken too far, we think, if it is interpreted to imply that research on human infancy should emphasize the dyad to the near exclusion of interest in the independent infant as an agent in its development.

There can be no doubt that social interaction plays a critical role in infant vocal learning and language acquisition; social learning allows us for example to acquire language-specific syllables, phonemic elements, and the largely arbitrary pairings of words with meanings in languages. But even deaf infants produce the same kinds of prelinguistic speech-like sounds, or "protophones" [30], as hearing infants in the first year of life [35]. Thus the importance of *hearing* speech sounds from the social environment does not appear to drive the initial development of protophones. In this paper, we seek to highlight the quantity of infant endogenous, non-cry vocal activity to further illuminate the role protophones play in supplying a basis for social learning.

Several studies have shown that dyadic vocal interaction increases the rate of protophone production (volubility), and the proportion of advanced vocal forms including canonical babbling appears to be particularly high during dyadic vocal interaction [5, 6, 8, 10, 27]. Yet surprisingly, the proportion of infant protophones that are social in nature has, to our knowledge, never been previously quantified, so the extent to which infant protophone production may be primarily social rather than endogenous is unknown.

## Intrinsic motivation to support vocal development

Intrinsic infant motivation for action and exploration has long been recognized. For example, Piaget's sensorimotor stage in the first two years of life is portrayed as a period wherein infants' self-generated gestures are produced without social intent, but rather for the pure enjoyment of experiencing sensorimotor activity [36, 37]. In anecdotal reports [38–42], the interpretation of this stage focused on the circular reactions of manual gestures, but Piaget did not emphasize circular reactions in the vocal domain [43].

The low level of focus on the infant as an independent agent of vocalization in prior research on development (see Appendix A of S1 Data) might be in part an unintended

consequence of the radical behaviorist tradition that for many decades treated behaviors as *responses* rather than *actions* [44, 45]. Panksepp and his colleagues have argued that we have not overcome the legacy of that radical behaviorism, and that even modern cognitive psychology continues to underplay the endogenous, emotion-driven actions of both humans and non-humans [46–49].

Breaking with the dominant tradition of infant development research, a role for intrinsic motivation as a primary mechanism to support vocal development has recently received increased attention [50–52]. In the Supplementary Material to a published article based on recordings made in our own laboratory [53], it was reported that infants across the first year of life produced the majority of their protophones when gaze was not directed toward another person. In a small-scale study from another laboratory with just 16 minutes of recording per infant at 6–8 months, infants produced more vocalizations when playing alone with toys than when engaged socially [54]. Another recent observational study found no significant difference in protophone volubility between a recording circumstance where parents talked to infants compared to circumstances where parents were in the same room and silent or not present in the room at all, suggesting that infants had an "independent inclination to vocalize spontaneously" in the absence of social interaction (p. 481) [7]. Importantly, the rate of protophone production has been reported to be very high, >4 protophones per minute during all-day audio recordings, across the entire first year, and even when infants were judged to be alone in a room, the rate was >3 per minute [55].

These findings suggest vocalizations are commonly produced endogenously. In other words, infants in these prior studies appear to have been intrinsically motivated to explore or practice sounds, in essence to play with sensorimotor aspects of sound production, although the evidence has been somewhat indirect. We propose that this vocal exploration may have a deeply significant role in vocal development, alongside the importance of caregiver-infant interaction and ambient language exposure. In spite of the possible importance of endogenous, exploratory vocalizations in language development, to our knowledge there is no published evidence specifically targeting the communicative function of infant protophones or the lack of it. Only with such work will it be possible to reliably quantify proportions of endogenous infant protophones and socially-directed ones. (see Appendix B of S1 Data, for information suggesting that both parents and non-parents tend to view infant vocalizations as being predominantly social rather than endogenous or exploratory).

We deem it important that such quantification be established in contexts with and without parent engagement across the first year of life. Prior studies suggest the proportions of endogenously-produced sounds may be high, but appropriate research requires direct comparison in different circumstances of potential interaction, especially when caregivers are attempting to interact with infants and when not. Providing such quantification may highlight the importance of endogenously generated vocalization and self-organization in prelinguistic vocal development [50, 52] and may help establish perspective about relative roles of endogenous and interactive factors in vocal development.

## Specific aims and hypothesis

Our primary goal is to determine the extent to which infants produce social and endogenous vocalizations at three ages and in two laboratory circumstances: An *Engaged* circumstance, where the parent attempts to interact with the infant, and an *Independent* circumstance, where the infant is present in a room, but the parent is interacting with another adult. This quantification is hoped to provide a standard against which we may be able to recognize the relative importance of infant protophones both as social and as endogenous. We hypothesize that

infants will produce predominantly socially-directed vocalizations in circumstances where parents are trying to interact with infants (*Engaged*) and predominantly endogenous vocalizations when parents are interacting with another adult while the baby is in the room (*Independent*).

## Materials and methods

Approval for the longitudinal research that produced data for this study was obtained from the IRB of the University of Memphis. Families were recruited from child-birth education classes and by word of mouth to parents or prospective parents of newborn infants. Interested families completed a detailed informed consent indicating their interest and willingness to participate in a longitudinal study on infant sounds and parent-child interaction.

We selected six parent-infant dyads (3 male, 3 female infants) from the University of Memphis Origin of Language Laboratory's (OLL) archives of audiovisual recordings. The dyads had been recorded while engaged in naturalistic interactions and play. The three female infants were initially selected for coding in an earlier study on imitation [56] which had utilized a coding methodology for judging illocutionary force similar to the one used in the present study. Three males were thereafter selected from the archives in order to balance the sample for gender. The selection was unbiased with regard to social vs. endogenous vocalization. All families lived in and around Memphis, Tennessee, and all but one infant were exposed to an English-only speaking environment (Infant 6 was exposed to English and Ukrainian at home). Parents were asked to speak English and no other language during the laboratory recordings. Criteria for inclusion of infant participants included a lack of impairments of hearing, vision, language, or other developmental disorders. Demographics and recording ages for each infant at each recording session are provided in Table 1.

### Laboratory recordings

Two laboratory recordings were selected from each of the 6 infants at approximately 3, 6, and 10 months, for a total of 36 sessions. The average session length was 19 minutes (range: 12–22 minutes). During recordings, the parent-infant pairs occupied a studio designed as a child playroom with toys and books. Laboratory staff operated four or eight pan-tilt video cameras located in the corners of the recording studio from an adjacent control room—there were three such recording laboratories at varying stages of the research. In all the laboratories, two channels of video were selected at each moment in time with the goal of recording: 1) a full view of the interaction or potential interaction, including the infant and any potential

**Table 1. Infant demographics.**

| Infant | Gender | Birth order | Maternal education | Home language | Age of recordings (months; weeks) | | | | | |
|--------|--------|-------------|--------------------|---------------|------|------|------|------|------|------|
| | | | | | 1 | 2 | 3 | 4 | 5 | 6 |
| 1 | F | 1 | PhD | English | 3;2 | 3;2 | 6;0 | 6;3 | 9;3 | 9;3 |
| 2 | M | 2 | BA | English | 4;2 | 4;2 | 6;0 | 7;2 | 11;2 | 11;2 |
| 3 | M | 1 | Some college | English | 3;2 | 3;2 | 5;0 | 6;0 | 10;0 | 10;0 |
| 4 | F | 1 | Some graduate school | English | 3;0 | 3;0 | 5;0 | 6;0 | 10;1 | 10;1 |
| 5 | M | 3 | Some college | English | 3;2 | 3;2 | 6;0 | 6;3 | 9;3 | 9;3 |
| 6 | F | 1 | PhD | English, Ukrainian | 4;0 | 4;1 | 6;0 | 7;0 | 11;3 | 11;3 |
| **Nominal age of recording** | | | | | **3 months** | | **6 months** | | **10 months** | |

All infants completed two recording sessions around 3, 6, and 10 months of age.

interactors (i.e., parent or laboratory staff) with one camera and 2) a close view of the infant's face with the other camera. Both the parent and the infant wore high fidelity wireless microphones, with the infant microphone <10 cm from the infant's mouth. Detailed descriptive information regarding the recording equipment can be found in previous studies from this laboratory [57, 58].

In roughly counterbalanced orders across ages, parents were either instructed to interact with the infant (the expected *Engaged* circumstance) or with another adult while the baby was in the room (the expected *Independent* circumstance). Later at the same age (usually on the same day), the dyad was recorded in the other circumstance. Parents were asked to interact with the infant and/or laboratory staff in a naturalistic manner. During the expected *Engaged* circumstance, parents were encouraged to engage in face-to-face interaction with the infant but were not restricted from interaction with others if someone came into the room (e.g., to adjust cameras, to answer parent questions, etc.). Similarly, in the expected *Independent* circumstance, parents were encouraged to keep their attention and interactive focus on the laboratory interviewer but were not restricted from engaging with the infants if they appeared uncomfortable or if the infants were repeatedly bidding for attention. The freedom allowed in these naturalistic recordings resulted in variation in the actual circumstance with respect to the expected circumstance. Our analysis took account of social directivity of infant utterances in the actual circumstances only.

### Coding for *Engaged* and *Independent* circumstances

As indicated above, the recordings had been intended to be differentiated neatly as primarily corresponding to *Engaged* or *Independent* circumstances, but the infants often sought attention from the parents during sessions intended by protocol to be *Independent*, or adults would engage in conversation with a staff member during sessions intended to be *Engaged*. For this reason, we re-categorized segments of time within each session in terms of whether they were actually *Engaged* or *Independent*. Pic 1 exemplifies this re-categorization.

These re-categorized segments were used in the analysis of the role of circumstance in the infant utterances. Table 2 shows the re-categorized, actual circumstance durations for each infant and infant age. (Appendix C of S1 Data) provides a more detailed breakdown of expected and actual circumstance durations for each infant and infant age.

The amount of time pertaining to the actual circumstances that occurred during the recordings varied substantially, including two periods of time that included so few utterances (< 5)

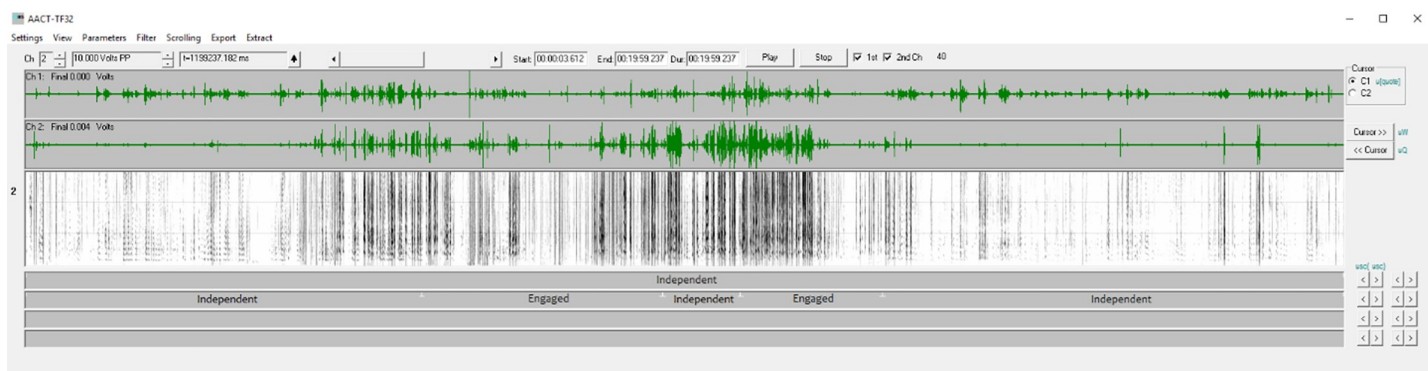

**Pic 1. Visualization of re-categorizing circumstance.** An example of one 20-minute recording (Infant 5 at 3 months) with the expected circumstance according to the protocol on line 1 of the coding field (below the spectrogram) and the re-categorization of actual circumstances on line 2. In this recording session, the parent was instructed to engage with the interviewer in accord with the *Independent* circumstance, but there were two substantial periods of time where the parent was actually directly engaged with the infant, and so those segments were re-coded as *Engaged*.

**Table 2. Actual circumstance durations.**

| Mean age | | 3 months | | 6 months | | 10 months | |
|---|---|---|---|---|---|---|---|
| Infant | Gender | Engd | Ind | Engd | Ind | Engd | Ind |
| 1 | F | 00:32:38 | 00:01:16 | 00:33:48 | 00:04:23 | 00:20:34 | 00:19:22 |
| 2 | M | 00:27:59 | 00:12:24 | 00:26:59 | 00:14:53 | 00:23:34 | 00:18:08 |
| 3 | M | 00:22:46 | 00:21:19 | 00:23:08 | 00:17:28 | 00:25:35 | 00:07:29 |
| 4 | F | 00:23:26 | 00:15:15 | 00:10:31 | 00:25:08 | 00:24:27 | 00:15:16 |
| 5 | M | 00:22:00 | 00:14:02 | 00:20:54 | 00:18:11 | 00:21:45 | 00:19:55 |
| 6 | F | 00:35:52 | 00:01:37 | 00:25:33 | 00:00:58 | 00:24:02 | 00:15:00 |

Duration of actual circumstance segments *Engaged* (Engd) and *Independent* (Ind) for each infant at each age. Overall, there were longer periods of time in the *Engaged* circumstance than in the *Independent* circumstance. The minimum duration was 00:58, maximum duration 32:52, with an average duration of 19:06.

we did not include them in the analyses, as indicated in the total protophone counts of Table 3. This substantial variation in circumstance duration, along with the variability of actual ages provided motivation for a statistical modeling approach that was robust and conservative with regard to such variations (see below).

## Coding of the function of infant protophones

Coding for circumstance, illocutionary function, and gaze direction was completed within the Action Analysis Coding and Training software (AACT) [59]. This coding software has been used and discussed extensively in previous research from this laboratory [25, 58, 60]. The software affords frame-accurate coordination of video and audio, which is displayed in a special version of the TF32 software [61]. TF32 includes both flexible waveform and spectrographic displays. Coders can view and listen with a scrolling audio display where a cursor indicates the location of the audio at each moment of playback. The utterances to be coded in the present work were labeled for vocal type and bounded in time for onsets and offsets in AACT in prior studies [53]. The AACT software allowed the coder to advance to each bounded utterance in turn for playback and coding in illocutionary force and gaze direction for the present study. The AACT software also allows users to export data that indicate whether an utterance occurred within an *Engaged* or *Independent* circumstance.

All infant protophones that had been previously bounded were also labeled for the present work in terms of *illocutionary force* [62–64] to indicate potentially communicative functions.

**Table 3. Protophone counts.**

| Mean age | | 3 months | | 6 months | | 10 months | |
|---|---|---|---|---|---|---|---|
| Infant | Gender | Engd | Ind | Engd | Ind | Engd | Ind |
| 1 | F | 446 | 4* | 310 | 47 | 182 | 118 |
| 2 | M | 230 | 202 | 181 | 122 | 108 | 70 |
| 3 | M | 311 | 163 | 158 | 102 | 133 | 81 |
| 4 | F | 273 | 227 | 103 | 384 | 233 | 138 |
| 5 | M | 328 | 257 | 330 | 147 | 89 | 117 |
| 6 | F | 442 | 13 | 381 | 4* | 116 | 107 |
| Average | | 338.33 | 144.33 | 243.83 | 134.33 | 143.5 | 105.17 |

Total counts of the number of protophones for the Engaged (Engd) and Independent (Ind) circumstances at each age for all infants. Cells marked with an asterisk (*) were excluded from analysis because they included fewer than 5 protophones.

Illocutionary force was originally defined by Austin as the social intention of a speech act, but has been extended in work in child development and animal communication to also encompass vocal acts produced with little or no social intention [53]. In this extended usage, vocal play, for example, is treated as an illocutionary force. Another example: a fussy protophone, not directed toward anyone, can be treated as having the illocutionary force of complaint.

Pre-linguistic infants express varying illocutionary forces and varying emotional content (i.e., positive, neutral, and negative) in early protophones beginning at birth [53, 65] (see Appendix D of S1 Data). This fact indicates that infants have the capacity to produce a single protophone type with different illocutionary forces on different occasions, indicating they possess a vocal capability that is, of course, required of all words and sentences in mature language. Put another way, infant protophones can be used with varying communicative intentions, for example, to gain attention, to continue vocal interaction when engaged with a caregiver, or to make a request. The same vocalization types can also be produced for the infant's own purposes when not engaged in social interaction at all, e.g., when vocalizing toward an object or when simply exploring sound for its own sake.

The determination of whether a vocalization is social or endogenous requires considering a variety of factors. One is gaze direction during infant vocalization, but another is the extent to which infants may bid for attention vocally even when they are not in the same room with caregivers. Judging directivity of infant vocalizations also requires taking into account the relative timing of infant and caregiver utterances as well as the content of utterances of adults who are present at the time of the recording, especially caregivers who presumably know a good deal about the capabilities of a particular infant. We make the assumption for this work that judgments about vocal directivity need to be made moment by moment, utterance by utterance, to account for the possibility that infants may engage and disengage in protoconversation. The judgments of the social or endogenous nature of infant protophones need to be made taking account of the broad context of events prior to and subsequent to each infant utterance, and factors such as timing, eye contact, perceived imitativeness, and meaningful responsivity must be allowed to yield intuitive judgments by the observer, where a balance among the factors provides the basis for the coding.

A coding scheme was created for making judgments on the illocutionary function of individual infant vocalizations in consideration of all of the above listed factors. **Social** protophones were labeled as such when, for example, the infant used them to initiate conversation, continue an ongoing interaction, imitate another person, or to complain or exult in a way that was directed to an adult as indicated by gaze, gestures, or other contextual factors. **Endogenous** protophones were identified as utterances infants produced for their own purposes; such events included vocal play, object-directed sounds, complaints and exultations not directed to others, or protophones with no clear illocutionary force. Brief descriptions of each code used for judgments of illocutionary function are provided in Table 4.

Our coding is founded on the assumption that human observers are naturally able to judge the extent to which vocalizations at any age are intended as communicative acts—otherwise how would humans know when to respond or participate in vocal engagement? If some parents are poor at making such judgments, they are surely at a disadvantage in child rearing, because they don't know when their infants are communicating or not. It makes sense that natural selection has produced parents (and potential parents) that are capable of recognizing when infants are communicating intentionally and when not. Consequently, the coding process takes advantage of natural capabilities of human observers and gauges the extent of their reliability by comparing agreement among observers.

During illocutionary coding, both the primary coder and an independent reliability coder took a broad view of each utterance and its context of production. The coding was conducted

**Table 4. Coding scheme for judgments of illocutionary function.**

| Endogenous vocalizations | | Social vocalizations | |
|---|---|---|---|
| *No Force* | Produced without obvious exploratory or social intention | *Call/Initiate* | Call or bid for attention directed toward another person |
| *Vocal Play* | Not directed to a person or object but apparently playful | *Continue* | Maintenance of a turn-taking sequence with another person with communicative intent |
| *Object-Directed* | Directed toward a toy or other object as indicated by body positioning, gaze, or gesture | *Imitation* | Matching of pitch or articulatory characteristics of another person's utterance while engaged in turn-taking |
| *Complaint* | Distress vocalization not directed to another person | *Complaint-Directed* | Distress vocalization directed to another person |
| *Exultation* | Celebratory vocalization not directed to another person | *Exultation-Directed* | Celebratory vocalization directed to another person |

Codes used for labeling illocutionary function of infant vocalizations. Contextual information such as gaze, body positioning, and timing was considered to make intuitive judgments on each infant utterance.

by watching the entire recording session. Then the coder started at the beginning of each session and observed everything that happened up to the point of each infant utterance, and then coded with repeat observation. That is, each time a protophone was located, the judgment of illocution was made based on the entire preceding context and the cursors could also be stretched so that, during repeated playbacks before coding for illocutionary force, the coder could, if necessary, see and hear the utterance plus a several-second context both before and after it repeatedly. If there was ambiguity about how to judge the possible social directivity of the utterance, the boundaries could be stretched further until the coder felt confident that no further stretching would improve the coding decision.

## Coding for gaze direction of infant protophones

Gaze direction coding was conducted independently of the illocutionary coding for all protophones and was based on gaze direction only. For this coding, sound was turned off, and the coder determined whether at any time during each utterance, the infant looked toward another person. The time frame of playback for the period during which the protophones occurred was expanded through a special setting in AACT by 50 ms before and 50 ms after the actual utterance boundaries as indicated based on the original protophone coding. This expansion of time frame for viewing was deemed important because of the low frame rate of video recording (~30 ms per frame) and ensured that the entire period of the vocalization was available for visual judgment. Utterances could be played repeatedly this way. They were judged as "directed to a person" (during any portion of the utterance plus or minus 50 ms) or "not directed to a person" (during the same period). For utterances that included no good camera view of the infant (the infant sometimes turned away from the selected cameras and vocalized before new cameras could be selected) or for utterances where the infant's eyes were closed, the coder indicated "can't see" or "eyes closed," respectively. The gaze direction analysis excluded all such utterances. A brief description of each code used for judgments of gaze direction is provided in Table 5.

## Coder training and coder agreement

For the coding in the present study, both the primary coder and the agreement coder were trained in infant vocalizations and illocutionary coding by the last two authors in a training sequence that has been described in several prior publications [25, 51, 53]. In brief, the training included 1) a series of 5 lectures on vocal development and coding of early vocalization and interaction, 2) an interleaved set of corresponding coding exercises using recorded data like

**Table 5. Coding scheme for judgments of gaze direction.**

| Directed Gaze | *Directed to Person* | Gaze clearly directed to another person's eyes or face |
|---|---|---|
| **Gaze Not Directed** | *Not Directed to Person* | Gaze clearly not directed toward another person |
| | *To Toy* | Gaze clearly directed toward a toy |
| | *To Mirror* | Gaze clearly directed into a mirror toward self or object in room and clearly not toward another person |
| **Unclear Gaze** | *Can't See* | Infant briefly outside of camera range; unable to make judgment |
| | *Eyes Closed* | Infant's eyes closed; gaze judgment not possible |
| | *Unspecified* | Gaze directed in the vicinity of person, unable to make a definitive judgment (e.g., too far away) |

Codes used for labeling directivity of infant gaze during vocalization. Each infant utterance was also coded for gaze to provide a secondary analysis on social directivity of protophone production.

that to be encountered in the current research; 3) comparisons of the outcomes of those coding exercises with regard to outcomes for other coders, with special reference to coder agreement and agreement with gold standard coding by the last author, who has been engaged in vocal development research for more than 40 years [66]; and 4) a certification process that resulted from reviews ensuring that coding results correlated highly with group coding and the gold standard coding and did not diverge from gold standard coding by more than 10% of mean values.

All the data of the present study were coded for illocutionary force (from which socially- and endogenous categories could be derived) by the first author, and approximately 30% of the total data set was coded independently for illocutionary force by the agreement coder. An original coding of gaze direction had been done on three of the six infants by a previous team of coders for the paper previously cited [53]. This completely independent prior coding on half of the data for the present study was available to offer an agreement check on the gaze coding done for the present paper.

## Results

### Protophone usage judged in terms of illocutionary functions

A total of 6,657 infant protophones were labeled across all 36 recordings (6 infants x 3 ages x 2 sessions). The data account for all infant utterances that were judged to be non-vegetative (burp, hiccough) and not fixed signals (cry, laugh) across the 36 laboratory recording sessions. Utterances where either gaze or illocution could not be judged were eliminated. Two segments were eliminated from analysis because of a very low number of protophones for that infant at that age in that condition (specifically, Infant 1, *Independent* at 3 months and Infant 6, *Engaged* at 6 months, see Table 3 in Methods). Only 8 protophones occurred in these 2 segments. We also limited the analysis to include utterances that could be judged based on audio and video both for illocutionary force and for gaze direction. The final set included 6,388 protophones.

To determine if the usage of endogenous protophones exceeded that of social protophones, we used *t*-tests comparing percentages of endogenous protophones against 50%. To test for effects of Age (3 levels) and recording Circumstance (*Engaged* vs. *Independent*), a different approach was required. We selected a logistic regression model based on Generalized Estimating Equations (GEE). GEE analyses are a non-parametric alternative to generalized linear mixed models that accounts for within-subject covariance when estimating population-averaged model parameters [67]. GEE is particularly appropriate for the data in question because

of the unequal amounts of data in the two circumstances and the lack of precise age matching across infants. GEE provides a conservative but robust method for such cases.

Fig 1 displays the overall percentages of protophones produced by the six infants across the two broad illocutionary groupings of endogenous and social. Infants used significantly more endogenous protophones across the three ages than social ones, with about 75% of all protophones being endogenous. By *t*-tests of the percentage of endogenous protophones, it was found they significantly ($p < .001$) exceeded 50% at all three ages. We found no notable change in the predominance of the endogenous protophones across Age, and indeed the GEE revealed no significant difference in the percentage of social protophones across Age ($p = 0.48$). A subsequent GEE analysis was conducted with Age as a continuous variable and produced the same pattern, with more endogenous protophones than social ones ($p < .0001$) and no Age effect ($p = .69$).

Similarly, *t*-tests of the proportion of endogenous protophones in the two circumstances (*Engaged* vs. *Independent*) showed that endogenous protophones significantly exceeded 50% in both circumstances ($p < .001$). Based on the GEE for data presented in Fig 2, infants used significantly more endogenous protophones in the *Independent* circumstance than the *Engaged* circumstance ($p < .03$). A separate GEE analysis in which only main effects were considered revealed a stronger Circumstance effect ($p < .0001$). The fact that endogenous protophones outnumbered social ones in the *Engaged* circumstance contradicted our hypothesis and highlighted the predominance of endogenous infant vocalization. A separate GEE analysis of the data treating Age as a continuous variable yielded similar results. Specifically, significant differences were seen for overall proportions of protophones between circumstances ($p < .001$) and non-significant differences across Ages ($p = .982$).

The pattern of results revealed by the illocutionary coding was similar for both the primary coder and the reliability coder, with 79% point-to-point inter-rater agreement on 30% of the recordings that were coded independently by the two observers. For both coders, endogenous protophones predominated, and the reliability coder—who had no knowledge of the

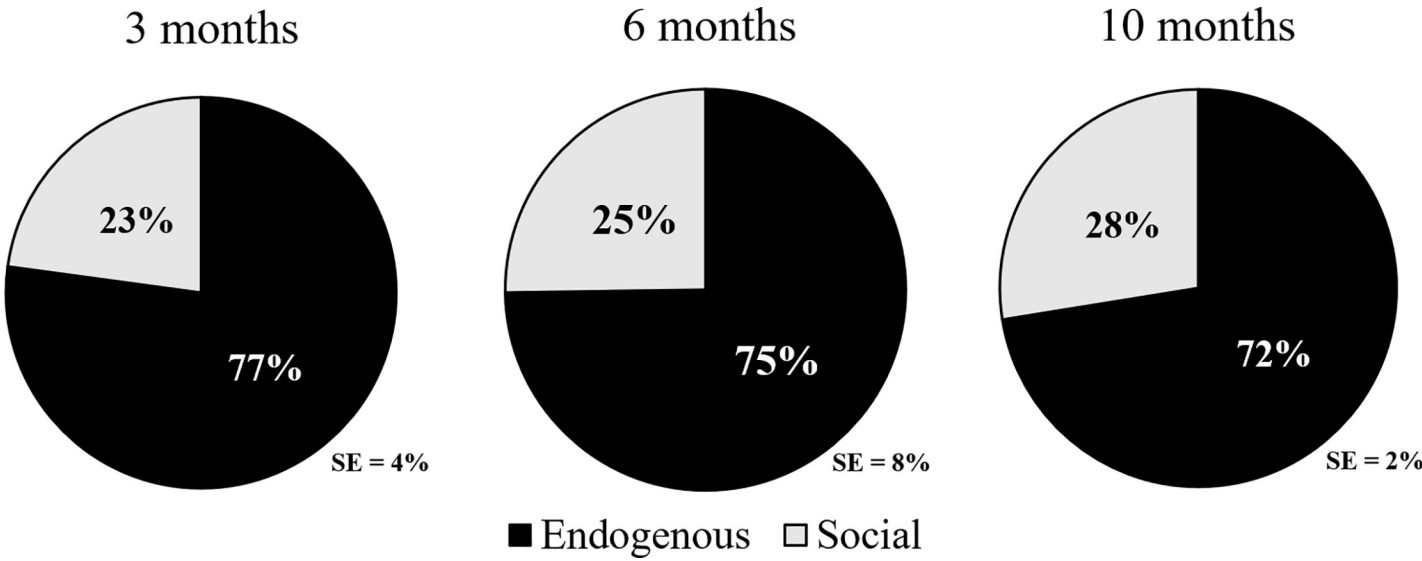

**Fig 1. Social and endogenous infant protophones across 3 ages.** Percentage of infant protophones that were judged to be endogenous (produced for the infants' own purposes) and social (overtly communicative) across all observations. Overall, infants primarily produced endogenous vocalizations (~75%), suggesting that the great majority of infant sounds are produced independent of social engagement in the first year. Furthermore, a non-significant main effect of Age is consistent with an interpretation of stable use of both social and endogenous protophones across the three ages.

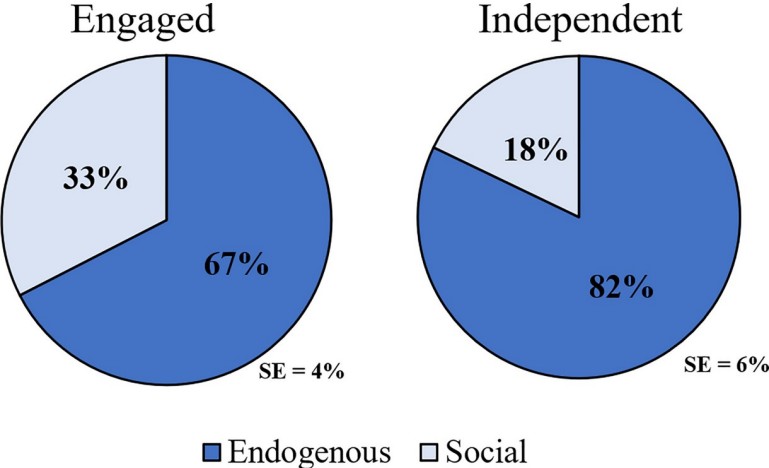

**Fig 2. Social and endogenous infant protophones across two circumstances.** Percentages of social and endogenous infant protophones across *Engaged* (parent and infant interacting) and *Independent* (parent and interviewer conversing while infant present in room) circumstances. Endogenous protophones predominated in both conditions.

hypotheses for this study—identified a slightly higher proportion of endogenous protophones (79.2%) than the primary coder (78.5%).

### Protophone usage based on gaze-direction judgments

As a check on the illocutionary coding, we considered an alternate, simpler way of gauging the function of infant protophones. The first author coded gaze direction during protophone production as being directed or not directed toward a person. Gaze judgments were made with sound off (video only) for all six infants.

Even though the function of protophones as determined by gaze-direction was not always the same as the function based on illocutionary judgments, the overall percentages of social protophones as determined by the two methods was very similar. That is, the great majority of infant protophones were judged to be produced with gaze directed somewhere other than towards any person in the room, just as the illocutionary judgments indicated the great majority of infant protophones to be endogenous. 72% of the infant protophones were deemed *not* to include person-directed gaze, while 75% were deemed endogenous by illocutionary coding.

In the earlier study mentioned above [53], 50% of the current sample had been coded for gaze direction, allowing for a robust analysis of independent inter-rater agreement. Inter-rater agreement on a point-to-point basis was 87% (of 3347 utterances). The results showed a strong predominance of protophones *not* being associated with gaze directed toward another person for both the earlier coders and the present one. Based on the same sample of utterances, the primary coder in this study found 64% of the utterances not to include person-directed gaze, while the previous (reliability) coder found 61% not to include person-directed gaze. These percentages represent only half the total sample (three of the six infants) and consisted heavily of samples from the *Engaged* circumstance; consequently, the percentages (64 and 61%) are lower than the 72% of utterances deemed not to include person-directed gaze for the whole sample as reported above.

Let us expand on why the gaze-direction and illocutionary coding methods do not yield exactly the same outcomes on the function of infant protophones. In the coding of illocutionary force, momentary gaze direction by the infant toward a person was sometimes not deemed to indicate the function of the vocalization. For example, a momentary glance directed to the

parent occasionally occurred even though the infant appeared to be engaged in vocal play. There were also a number of cases where the coder deemed a protophone to be social in illocutionary coding, even though gaze direction toward a person was deemed absent. Such cases often corresponded to interactional sequences where the relative timing of utterances suggested the infant was engaged and directing the protophone to the parent, even though the infant was looking away.

## Discussion

Overall, infants used about three times as many endogenous protophones as social ones. This predominance remained stable across the three ages. Even in the *Engaged* circumstance, where parents were trying to engage with their infants, endogenous protophones predominated, with twice as many judged to be endogenous as social. In the *Independent* circumstance, where parents were engaged in conversation with laboratory staff, the endogenous protophones predominated to a substantially greater extent, with four times as many endogenous as social.

The low rate of socially-directed vocalizations of infants in the first 10 months as reported here has required us to reorient our thinking about the functions of infant protophones. It seems important to draw attention to the fact that for all the sessions of recording reported on here the caregivers and infants were in the same room, and caregivers were aware that they were being recorded. The caregivers also knew the study was about vocal development, and it was assumed they would endeavor to elicit infant vocalization and thus interact as much as possible. They often attended to infant vocalizations even in the designated *Independent* circumstances, sometimes responding to infant protophones with infant-directed speech (IDS), a pattern of caregiver responsivity that required some restructuring of our analysis to assign segments within sessions appropriately to the actual *Engaged* and *Independent* circumstances. Consequently, we presume parents tried to maximize their infants' socially-directed vocalization—and yet the rate was low.

Partly because the *Independent* circumstance resulted in a considerably larger predominance of the endogenous protophones than the *Engaged* circumstance, we presume that even more naturalistic recordings might produce an even greater predominance of endogenous protophones. That is, we suspect that the percentage of infant protophones that are socially directed in the natural environment of the home could be considerably *lower* than the values estimated here. This suspicion is supported by recent results where we compared the amount of IDS occurring in laboratory recordings for 12 infants (three of whom are among those represented in the present work) to the amount of IDS occurring in all-day LENA recordings [68] conducted in the home with the very same infants at approximately the same ages across the first year of life [51]. IDS was six times more frequent in the laboratory recordings than in randomly-selected five-minute samples from the all-day recordings when infants were awake. Thus, we reason that the percentage of endogenous protophones at home could be considerably higher than we have seen in the present work, since IDS is considerably lower. We plan to explore the rate of endogenous vocalization in all-day recordings in subsequent efforts. We also aim to study a larger sample of infants and to consider more differentiated circumstances of recording.

Our results contradict expectations that have often been apparent in the field of child development, where infant vocalizations are generally treated as responses to adult utterances or as attempts to engage adults in social interaction or to seek help from adults. Why has there been relatively low emphasis on exploratory or endogenous vocalization? It seems likely that the answer lies in the amount of attention given by caregivers to infant vocalizations that are directed toward them as opposed to those that are not. We assume parents and other

caregivers notice and remember vocalizations that appear to be social in nature to a greater extent than endogenous ones, and perhaps developmental researchers are similarly influenced by the salience of infant sounds that are embedded in protoconversation. Furthermore, parents may attend to any unique type of spontaneously produced protophone—irrespective of the communicative intent—and adapt their behavior to promote continued production of that particular sound, creating the appearance of, or perhaps initiating engagement with the infant. Indeed, we have reported evidence suggesting caregivers pay greatest attention to salient vocal signals such as those occurring in imitation, even though vocal imitation is surprisingly rare in the first year [69]. Caregivers, and thus people in general, may be inclined to overestimate the proportion of salient vocal signals such as imitation or immediate responses in protoconversation since it seems likely these are the sounds to which parents attend the most. So when they render estimates, they tend to overstate the frequency of occurrence of the social ones. It is only with systematic counting of every vocalization occurring in recorded samples, as has been done in the present work, that it becomes possible to determine that the great majority of infant protophones are in fact directed to nobody.

The results strongly suggest, then, that babies vocalize predominantly for their own endogenous purposes, hundreds or even thousands of times daily—4–5 times per minute of wakeful time based on randomly-sampled segments from all-day recordings at home [51]. There is considerable evidence that not just in vocalization, but in other realms as well, babies are not passive learners and in fact regularly influence their own experiences [70]. A fundamental question that requires answering based on the present work is: If protophones are not directed to caregivers, what is their purpose from a developmental or an evolutionary standpoint? What advantage could be associated with producing vocal sounds that are largely affectively neutral, produced most commonly in apparent comfort, but without social directivity [53, 65]?

One possibility is that infants may be learning the range of capabilities of their vocal system through sensorimotor exploration. We see evidence of this possibility when infants produce squeals for extended periods, repeatedly make small whisper sounds or raspberries, or babble the same syllables repeatedly to a toy. Of course it seems likely that endogenous and social vocalization both contribute to the development of the speech system [37, 43]. But importantly, the sounds infants use in endogenous vocal activity provide the raw vocal material that parents are able to use in engaging their infant in protoconversation.

Members of our research group and John L. Locke have argued elsewhere [63, 71–73] from an evolutionary-developmental (evo-devo) perspective [2, 4, 74, 75] that high rates of endogenous infant vocalization and vocal play may constitute fitness signals. The idea is based on the fact that the human infant is altricial (born relatively helpless) and has a long road ahead of requiring caregiver assistance for survival—the need for such caregiving lasts literally twice as long as in our closest ape relatives [76]. Consequently, we have argued that the human infant experiences selection pressure on the provision of fitness signals that could have the effect of eliciting long-term investment from caregivers, whose evolutionary goal can be portrayed as perpetuation of their own genes through grandchildren. From this point of view, caregivers should invest more in infants who seem healthy and tend to neglect infants who seem less healthy. We operate under the assumption that the production of comfortable vocalization can signal well-being and good health. This pattern of fitness signaling is hypothesized to have applied to the ancient hominin infant, who has been presumed in accord with the hominin "obstetrical dilemma" [77], to have been more altricial than other apes as soon as humans were bipedal. In accord with the reasoning about bipedality—which proves surprisingly difficult to confirm in the fossil record [78, 79]—bipedality had narrowed the human pelvis and required the hominin infant to be born with a smaller head and brain and thus to be more altricial than other apes. While the roots of human vocal flexibility appear to lie in their value as fitness

signals in a distant hominin past, modern human infants are not less altricial than their distant forebears, and consequently we reason that endogenous protophones continue to be under selection pressure as fitness signals in human infancy.

One might ask, if fitness signaling is the primary advantage of protophones, why do infants not endeavor to direct their protophones primarily toward potential caregivers? Of course, some of the time they do, as indicated by our data. When they do not, the protophones may still be heard and noticed, if only semi-consciously by potential caregivers. A parent may hear comfortable infant protophones and draw the unspoken conclusion that the infant is well and needs no immediate attention. Regular events of noticing the infant's well-being may reinforce a caregiver's commitment to long-term investment precisely because it suggests that particular infant is healthy and thus likely to be a good investment for survival and reproduction. So it may pay for the human infant to produce protophones at prodigious rates in case someone might be listening.

The production of protophones in infancy at the beginning of the communicative split between ancient hominins and their ape relatives, perhaps millions of years ago, seems likely to have laid a foundation for a more extensive use of vocalization as a fitness signal later in life, for example, in mating or in alliance formation [72]. And as the amount of protophone-like vocalization became more well-established in the hominin line, it surely provided a foundation for more elaborate uses of vocalization, ratcheting from simple fitness signaling toward more and more language-like uses [63].

Play is widely recognized as a theater for practice of the behaviors young mammals will need as they proceed through life [80, 81]. But it is important to note that playful behavior can serve not only as practice, but also as a fitness signal for the altricial young of many species. Our suggestion is that protophones can be seen (in the substantial majority of cases) as playful indicators of well-being, but they would seem to contribute at the same time to a sort of preparation for the future in mating, in alliance formation, and ultimately (nowadays) in the development of language.

## Supporting information

**S1 Data.**
(DOCX)

## Acknowledgments

We wish to thank the families in Memphis whose infants participated in this research, the graduate student reliability coders, and the survey participants (Appendix B of S1 Data Opinion Study).

## Author Contributions

**Conceptualization:** Helen L. Long, Dale D. Bowman, Hyunjoo Yoo, Megan M. Burkhardt-Reed, D. Kimbrough Oller.

**Data curation:** Helen L. Long, Hyunjoo Yoo, Megan M. Burkhardt-Reed, Edina R. Bene, D. Kimbrough Oller.

**Formal analysis:** Helen L. Long, Dale D. Bowman, D. Kimbrough Oller.

**Funding acquisition:** Edina R. Bene, D. Kimbrough Oller.

**Investigation:** Helen L. Long, D. Kimbrough Oller.

**Methodology:** Helen L. Long, Megan M. Burkhardt-Reed, D. Kimbrough Oller.

**Project administration:** Helen L. Long, Edina R. Bene, D. Kimbrough Oller.

**Resources:** Hyunjoo Yoo, Edina R. Bene, D. Kimbrough Oller.

**Software:** Hyunjoo Yoo, Megan M. Burkhardt-Reed, Edina R. Bene, D. Kimbrough Oller.

**Supervision:** Helen L. Long, Edina R. Bene, D. Kimbrough Oller.

**Validation:** Helen L. Long, D. Kimbrough Oller.

**Visualization:** Helen L. Long, D. Kimbrough Oller.

**Writing – original draft:** Helen L. Long, D. Kimbrough Oller.

**Writing – review & editing:** Helen L. Long, Dale D. Bowman, Hyunjoo Yoo, Megan M. Burkhardt-Reed, Edina R. Bene, D. Kimbrough Oller.

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
