## [Decision Letter · Decision Letter 0]

7 Apr 2020

PONE-D-19-29493

Social and non-social functions of infant vocalizations

PLOS ONE

Dear Ms. Long,

Thank you for submitting your manuscript to PLOS ONE. After careful consideration, we feel that it has merit but does not fully meet PLOS ONE’s publication criteria as it currently stands. Therefore, we invite you to submit a revised version of the manuscript that addresses the points raised during the review process.

As you will read in the attached reviews, both reviewers support the publication of the manuscript. Yet they both have some questions and comments and concerns about the paper which should be addressed in the revision.

These can be summarised as follows:

1) Background literature and theoretical assumptions. Both reviewers felt that the paper takes a somewhat narrow point of departure. Reviewer 1 for example suggests a more comprehensive literature review in the introduction. Along the same lines Reviewer 2 suggests that the authors adopt a rather narrow theoretical view. Both reviewers suggest that, both in the introduction and the discussion, there is a need to acknowledge other existing theoretical approaches and different interpretations of the findings.

2) Concerns with the Survey data. Both reviewers raised some concerns about the setup of the study and the contribution of its findings to the paper. One of the reviewers suggests minimising or excluding this study. They have provided questions and comments that should be addressed both in the manuscript as well as in the reply to reviewers.

3) Some questions and comments on data coding, clarification of selection criteria and further comments on the analysis (such as e.g. grouping ages instead of using age as a continuous variable). There are also some comments on potential additional analyses.

4) Data availability. Review 1 commented on potentially making available annotated data if the raw data cannot be made available. Could the authors please also respond to this comment?

I believe that the reviewers have provided constructive comments which will allow the authors to adapt the manuscript and respond.

We would appreciate receiving your revised manuscript by May 22 2020 11:59PM. To enhance the reproducibility of your results, we recommend that if applicable you deposit your laboratory protocols in protocols.io, where a protocol can be assigned its own identifier (DOI) such that it can be cited independently in the future. For instructions see: http://journals.plos.org/plosone/s/submission-guidelines#loc-laboratory-protocols

We look forward to receiving your revised manuscript.

Kind regards,

Iris Nomikou, Ph.D.

Academic Editor

PLOS ONE

Journal Requirements:

Additional Editor Comments (if provided):

Dear authors,

I would like to apologise for the long wait. I took a lot of time to secure the reviewers. It was my mistake to grant reviewers extension after extension in hope that they would submit their reviews. This was because I really wanted to secure the most appropriate reviewers for your paper, which I think in the end I did.

As you will see both reviewers are really positive and so I hope that the second round of the review will run smoothly.

Best wishes,

Iris Nomikou

Reviewers' comments:

Reviewer's Responses to Questions

**Comments to the Author**

1. Is the manuscript technically sound, and do the data support the conclusions?

Reviewer #1: Partly

Reviewer #2: Yes

2. Has the statistical analysis been performed appropriately and rigorously? 

Reviewer #1: Yes

Reviewer #2: Yes

3. Have the authors made all data underlying the findings in their manuscript fully available?

Reviewer #1: No

Reviewer #2: No

4. Is the manuscript presented in an intelligible fashion and written in standard English?

Reviewer #1: Yes

Reviewer #2: Yes

5. Review Comments to the Author

Reviewer #1: The manuscript is technically sound but two key changes are necessary for the conclusions to be directly tested/supported.

First, it is stated that a social interpretation is predominant in the literature, but only one citation is provided. I recommend a small but unbiased literature review (perhaps sampling 50 papers from pubmed with search terms "infant vocalization protophones"), to make sure that "research on human infancy [emphasizes] the dyad to the near exclusion of interest in the independent infant as an agent in its development".

On a related note, it would be important to clarify exactly the instructions provided in the MTurk questionnaire; is there a way to be certain that respondents had the same definition of 'social' as your coders? How would someone understand "non-socially directed"? Perhaps to be even more certain, the questionnaire could be repeated with other ages (babies, children, adults), to establish that people are not just answering towards the middle of the scale.

Second, in the discussion, the only interpretation that is put forward is also based on a social/dyadic view of child vocalizations. Why would infant production not be also internally motivated? One could make the case that crawling, walking, mouthing objects are all also fitness signals; but it seems equally plausible that these are all behaviors exhibited by a young learner who is exploring the physics of this world and his/her control of his/her own body. Moreover, it is unclear to what extent the fitness explanation of babbling may not be culturally specific (Brazelton, 1972).

On an unrelated note, given that there is so much variation in the duration of sessions, please consider controlling for that in a supplementary analysis (which could be in an appendix).

DATA AVAILABILITY

The authors state they cannot share the recordings because parents did not okay this. However, the annotations should be shareable since they are fully deidentifiable. At the very least, the authors should share a table like Table 3 but splitting socially directed versus not, gaze-directed or not.

MINOR:

- p. 8 "to estimate the percentage of how many of these sounds" were these literally the instructions? Because this language seems confusing to me (either percentage or how many, not "percentage of how many")

- p. 8 line 166 for a total of 3 judgments or 9 judgments?

- p. 9 What were the attention checks? Please check order of responses in Table 1 for frequency around children

- p. 10-13 There are several mentions of selection from a larger data set (more children, more recordings). Can you please clarify how you made sure these selections were not biased?

p. 14 lines 258+ But doesn't this suggest that you may be over-estimating social vocalizations? I'm also uncertain how to interpret the comment on p. 22, line 434-5

p. 15 line 275-6 Please specify what other contextual factors. Timing is mentioned in the discussion; what else could be used? And for timing, how was timing integrated?

p. 17 line 321 "and many more"  ", particularly" (to avoid a reading in which the first half of the sentence refers to younger babies)

p. 18 line 344 "resulting"  "remaining"

p. 22 line 439 Do you really mean "suspicious"?

Figure captions: please provide short explanations of "interactive" and "socially directed"

REFERENCE

Brazelton, T. B. (1972). Implications of infant development among the Mayan Indians of Mexico. Human development, 15(2), 90-111.

Reviewer #2: This paper presents data, first, from a survey of public opinion on the social orientation of infant vocalization in the first year of life and, second, from multiple recordings of infants with their parents in a laboratory study at various ages across the first year. The findings show that contrary to both common belief and dominant scientific theory, infant vocalization is more often nonsocial than directed to another person. The view put forth in the discussion section is that the primary function of preverbal vocalization is not to communicate internal states or messages but rather to signal fitness for developmental adaptation and change. Infant vocalization is thus presented as a residual behavior from the long phylogenetic process of human evolution and as an adaptation to the constraints brought about by altriciality.

Overall the paper is well-written and well-presented, its studies seem rigorously conducted and their findings are reported with sufficient clarity. The main finding, that infants produce many more nonsocial vocalizations both when they are interacting with a parent and when merely in the presence of their parent, is important, novel and challenging. The study provides evidence that, from the cooing to the babbling stages of vocal development, infants mostly partake in vocal play that is not directed toward another person. The evidence is based on naturalistic longitudinal recordings of 6 infants and their parents made by a team of researchers in their University of Memphis laboratory. Although the number of participants is small, the evidence is based on a large tally of individual infant vocalizations, or protophones, identified from multiple segments compiled together for each of two circumstances, interactive and non-interactive. The data is obtained from the audio and video segments. Thus 6649 protophones are categorized by coders based on two types of coding strategies. Each protophone is coded as social or nonsocial, first according to its ‘illocutionary force’ and, second based on whether or not it coincides with gaze oriented to the parent. The first coding strategy is based a subjective appraisal of the context within which the protophone was produced (over a few seconds) and the second is based on a micro-analysis of video footage without sound corresponding to the protophone + or – 50 ms.

Though it is fairly usual in studies using naturalistic data for researchers to reorganize the original data in a way that makes it relevant for the study, the description of the pertinent segments in the methods sections could be made clearer. Firstly, table 2 presents the exact ages at which recordings in both circumstances were made for each of the 6 infants and shows there are 6 recording sessions per infant (with age and condition as within subject factors). The ages range from 3 months to 11;3 months but are grouped into 3 age categories (3, 6 and 10 months). Although there is a clear gap in the data between 7;3 and 9;3 months, there is no obvious reason to distinguish groups between the 3- and 6-month categories. Indeed, much research, mostly by the last author of this paper, has shown fine-grain developmental change in the qualities of infant vocalization (for example, solitary vocal play increasing between 4 and 5 months of age compared to 2 and 3 months of age). I would recommend considering age as continuous, sequential time points rather than as a categorical factor. Secondly, table 3 shows how the data had to be recompiled based on identification of segments described as interactive and non interactive circumstances. The authors explain on page 11 that when for example an infant in the non interactive condition (parent talking to another adult), “sought attention from the parents” the segment was re-labelled as ‘interactive’. They also explain that in the interactive sessions the experimenters could engage in conversation with parents and in such a case the segment would also be re-labelled as ‘non interactive’. However, the specific criteria for re-labelling of segments is not clear. What do the authors mean by “sought attention”? Why were parents ever given the opportunity to interrupt interaction with the infant in the “interactive sessions” and in the case of an interruption shouldn’t the whole segment be excluded from analysis?

By and large I am not convinced by the relevance or usefulness of the survey study. I do not understand its purpose and I feel the paper would be stronger without it. It is fairly obvious that random groups of adults would remember only the vocalizations that are efficient in gaining attention from adults. Even parents involved with their infants on a daily basis often fail to hear the noncry vocalizations they produce. Indeed, even researchers accustomed to careful listening fail to notice some vocalizations based on audio recordings and require visual plots in order to appreciate both quantity and quality of infant sound making (I speak from experience here!). The authors seem to want to show that researchers who have focused their attention on specific brief episodes where infants and adults are socially engaged provide a biased perspective on infant vocalization because they select out most everyday sounds produced by infants. And it is quite convincing enough to state this without recourse to Amazon mTurks. With regard to the design of the survey I am concerned that the wording of the questions entails some bias linked to the use of contrastive questions involving negation (socially-directed vs. NOT directed) rather than two confirmations (e.g. socially-directed and for own enjoyment).

I am also concerned, at a more theoretical level, that the authors have a rather narrow view of what some researchers mean by the term ‘interaction’ between infants and social partners. The authors claim that the large proportion of endogenous, nonsocial or “intrinsically motivated” vocalization they find for all infants at all ages shows that infants are active vocal learners, whereas research showing high rates of social communicative vocalization entails a view of the infant as a passive recipient of vocal input. However, much research on vocal interaction presents a very different perspective, one where the infant on the contrary is involved in a socially-motivated agency which is crucial for any form of interpersonal coordination, intersubjectivity or attunement (see research and positions of Trevarthen, Beebe, Stern). The term “intrinsic motivation” has in fact been used in infant development studies for decades by Trevarthen to refer to the infant’s innate motives for social engagement with other persons. The view that infants learn through active solitary exploration is rooted in a Piagetian approach and has come under serious criticism in infant psychology. There has recently been renewed interest in the role of social interaction involving contingent vocalization between adults and infants (such as reflected in studies by Goldstein and collaborators, Hirsh-Pasek and Golinkoff or Pat Kuhl and others) for language acquisition. However, the emphasis of these researchers on social contingency entails a view of interaction as a mechanistic social feedback loop where infants learn responses to cues. Careful descriptive and data-drive research on selected episodes of attentive adult-infant engagement has shown that infants are remarkably active when engaged with another person and explore more varied sound forms when scaffolded by imitative vocalization of their parent (Gratier & Devouche, 2011). It does not seem in any way incompatible that infants may learn both though interactively scaffolded vocal expression and through active solitary exploration of sound-making. In this sense, this study indeed provides invaluable insight into the extent to which infants may engage in vocal play that is not directed to others. But perhaps here quantity is not the crux of the matter, perhaps those brief moments of coordinated engagement between parents or other adults and infants provide as much (or more) developmental gain in less time than longer periods of endogenous self-motivated sound making.

It is indeed intriguing that the authors find that most protophones produced even in the interactive circumstances are nonsocial. The authors assume that a vocalization is social only if it is associated with gaze or other social cues. However, the relative timing of vocalization and of other social cues merit perhaps more detailed analysis than is provided by the coding in this paper. It is plausible that infants rely on multi-modal timing so that they may gaze or smile at a partner not while they vocalize but shortly before or after vocalization. This finding also suggests that even when parents are attentive to their infants and stimulate their engagement through infant-directed speech, mutual coordinated engagement is rare and brief. But again, however brief, it may be highly adaptive and of great importance to vocal development and to language acquisition and cognitive development.

The authors ask an important question about their findings. What is the purpose of undirected preverbal vocalization? The explanation provided by the authors (protophone as fitness signal) is thought-provoking but excludes other plausible evolutionary explanations that merit attention. Dean Falk for example has developed a theory that infant-directed speech, evolved in order for parents to maintain continued contact with infants incapable of clinging on to them, gave rise to elaborate vocal interaction paving the way to language. Ellen Dissanayake suggests that the temporal sequencing ability that evolved from multimodal caring interactions between mothers and infants was a crucial adaptation for language and for cultural practices such as ritual and art. It would also be worth discussing to what extent the activity of sound-making is associated with a creative, intelligent and prospective process in infant development rather than acting as mere beeps of a survival instinct. Finally, it is not clear how the fitness signaling function of protophone production prepares infants for the major incremental shifts involved in learning to produce language.

In conclusion, this paper provides exciting and thought-provoking findings that deserve to be published and that should stimulate further research on the important questions it raises. My main recommendation is to discard or minimize the opinion survey and to provide some clarification of the selection criteria of the coded segments. I also hope some of the points I have raised regarding the question of agency and vocal learning in social engagement might make their way into the discussion section.

6. PLOS authors have the option to publish the peer review history of their article (what does this mean?). If published, this will include your full peer review and any attached files.

Reviewer #1: Yes: Alejandrina Cristia

Reviewer #2: Yes: Maya Gratier

---

## [Author Response · Author response to Decision Letter 0]

28 May 2020

See attached document "Response to Reviewers.docx" for comments to each reviewer.

---

## [Decision Letter · Decision Letter 1]

30 Jun 2020

PONE-D-19-29493R1

Social and endogenous infant vocalizations

PLOS ONE

Dear Dr. Long,

Thank you for submitting your manuscript to PLOS ONE. I have now received both reviews on your manuscript. I support the reviewers' view that the manuscript has improved and meets the requirements for publication. 

I am giving a decision of Minor Revision because the data can still not be accessed on the OSF website. Both myself and Alejandrina Cristia attempted to visit the page and it was restricted. I agree with the Reviewer that it is good to have the data checked by an external person. Also the reviewer suggested a final proofread for typos.

Could you please make the OSF website public and communicate this to me together with a submitted proofread manuscript by Aug 14 2020 11:59PM ? If you will need more time than this, please reply to this message or contact the journal office at plosone@plos.org. Please include the following items when submitting your revised manuscript:

We look forward to receiving your revised manuscript.

Kind regards,

Iris Nomikou, Ph.D.

Academic Editor

PLOS ONE

Reviewers' comments:

Reviewer's Responses to Questions

**Comments to the Author**

1. If the authors have adequately addressed your comments raised in a previous round of review and you feel that this manuscript is now acceptable for publication, you may indicate that here to bypass the “Comments to the Author” section, enter your conflict of interest statement in the “Confidential to Editor” section, and submit your "Accept" recommendation.

Reviewer #1: All comments have been addressed

Reviewer #2: All comments have been addressed

2. Is the manuscript technically sound, and do the data support the conclusions?

Reviewer #1: Yes

Reviewer #2: Yes

3. Has the statistical analysis been performed appropriately and rigorously? 

Reviewer #1: Yes

Reviewer #2: Yes

4. Have the authors made all data underlying the findings in their manuscript fully available?

Reviewer #1: Yes

Reviewer #2: Yes

5. Is the manuscript presented in an intelligible fashion and written in standard English?

Reviewer #1: Yes

Reviewer #2: Yes

6. Review Comments to the Author

Reviewer #1: The revised manuscript is clear, makes good contact with prior literature, and provides all details necessary for a reader to make their own mind regarding the results, while still presenting the authors' preferred interpretations as such.

Please note that the OSF component is not yet public, so I was not able to check the data. I would be happy to do so when the link is rendered public, since it is always useful to have an outsider check that e.g. column names are understandable.

Also, please double check the manuscript for typos. The versions that were uploaded were the old one + the one with tracked changes, which I read for my review. So I'm not sure if there may be typos in the final version (without tracked changes).

Reviewer #2: (No Response)

7. PLOS authors have the option to publish the peer review history of their article (what does this mean?). If published, this will include your full peer review and any attached files.

Reviewer #1: **Yes: **Alejandrina Cristia

Reviewer #2: **Yes: **Maya Gratier

---

## [Author Response · Author response to Decision Letter 1]

1 Jul 2020

Replies to comments regarding returned manuscript:

1) The OSF data are now publicly accessible at the link previously provided: https://osf.io/hb834/

2) The manuscript and supplementary material have been reviewed for typos and errors.

3) We wish to opt out of supplying lab protocols to an online repository due to the evolving nature of the methodologies and coding schemes in studies out of this lab.

---

## [Decision Letter · Decision Letter 2]

21 Jul 2020

Social and endogenous infant vocalizations

PONE-D-19-29493R2

Dear Dr. Long,

We’re pleased to inform you that your manuscript has been judged scientifically suitable for publication and will be formally accepted for publication once it meets all outstanding technical requirements.

Kind regards,

Iris Nomikou, Ph.D.

Academic Editor

PLOS ONE

Additional Editor Comments (optional):

In your data repository, please make sure to add a definition of the column headers and field properties as proposed by the reviewer below.

Thank you for you collaboration in this review process.

Reviewers' comments:

Reviewer's Responses to Questions

**Comments to the Author**

1. If the authors have adequately addressed your comments raised in a previous round of review and you feel that this manuscript is now acceptable for publication, you may indicate that here to bypass the “Comments to the Author” section, enter your conflict of interest statement in the “Confidential to Editor” section, and submit your "Accept" recommendation.

Reviewer #1: (No Response)

2. Is the manuscript technically sound, and do the data support the conclusions?

Reviewer #1: (No Response)

3. Has the statistical analysis been performed appropriately and rigorously? 

Reviewer #1: (No Response)

4. Have the authors made all data underlying the findings in their manuscript fully available?

Reviewer #1: (No Response)

5. Is the manuscript presented in an intelligible fashion and written in standard English?

Reviewer #1: (No Response)

6. Review Comments to the Author

Reviewer #1: Thanks for sharing this, on behalf of the community!

Please add a sheet or a separate file that explains the column headers and field properties:

- ageexact - unit (months?)

- agecategory - unit (months)

- meaning of the categories in Circumstance IllocutionCode IllocutionCategory GazeCode

7. PLOS authors have the option to publish the peer review history of their article (what does this mean?). If published, this will include your full peer review and any attached files.

Reviewer #1: No

---

## [Editor Report · Acceptance letter]

23 Jul 2020

PONE-D-19-29493R2 

Social and endogenous infant vocalizations 

Dear Dr. Long:

I'm pleased to inform you that your manuscript has been deemed suitable for publication in PLOS ONE. Congratulations! Your manuscript is now with our production department. 

Kind regards, 

on behalf of

Dr. Iris Nomikou 

Academic Editor

PLOS ONE